# Neural Network Surgery: Combining Training with Topology Optimization

## Abstract

With ever increasing computational capacities, neural networks become more and more proficient at solving complex tasks. However, picking a sufficiently good network topology usually relies on expert human knowledge. Neural architecture search aims to reduce the extent of expertise that is needed. Modern architecture search techniques often rely on immense computational power, or apply trained meta-controllers for decision making. We develop a framework for a genetic algorithm that is both computationally cheap and makes decisions based on mathematical criteria rather than trained parameters. It is a hybrid approach that fuses training and topology optimization together into one process. Structural modifications that are performed include adding or removing layers of neurons, with some re-training applied to make up for incurred change in input-output behaviour. Our ansatz is tested on both the SVHN and (augmented) CIFAR-10 datasets with limited computational overhead compared to training only the baseline. This algorithm can achieve a significant increase in accuracy (as compared to a fully trained baseline), rescue insufficient topologies that in their current state are only able to learn to a limited extent, and dynamically reduce network size without loss in achieved accuracy.

## 1 Introduction

A common problem for any given machine learning task to be addressed with artificial neural networks (ANNs) is how to choose a sufficiently good network topology. Picking one that is too small may not yield acceptable prediction accuracy. To improve results, one can keep adding structural elements to the network until the desired accuracy value has been reached. However, at the same time too large networks may cause an explosion in computational cost for both training and evaluation. Where the sweet spot in between lies is unclear, and heavily dependent on the given task. A priori optimization is not easily possible, since reliable estimates on network behaviour already require training results, and also there exists no generalization for which topology will fit which problem. Researchers have applied a number of search strategies such as random search (Li & Talwalkar, 2019), Bayesian optimization (Kandasamy et al., 2018), reinforcement learning (Zoph & Le, 2017), and gradient-based methods (Dong & Yang, 2019). Another technique applied since at least Miller et al. (1989) are so called (neuro-) evolutionary algorithms. These algorithms serve to evolve the network architecture, often also training network weights at the same time (Elsken et al., 2019).

In this paper we propose a novel training regime incorporating a genetic algorithm that reduces computational cost compared to state of the art approaches of this kind (Dong & Yang, 2019; Li & Talwalkar, 2019). We achieve this by re-using network weights for competing modification candidates instead of retraining each net from scratch, branching off modification candidates during training, and letting them compete against each other until a new main branch is selected. This better fuses the evolutionary optimization paradigm with the ANN training into an integrated framework that folds both processes into a single training/topology optimization hybrid. As such, evolutionary steps are not carried out by a meta-controller or other black-box-like implementations. We make use of mathematical tools such as singular value decomposition (SVD) and the Bayesian information criterion (BIC) (Schwarz, 1978) for network weight analysis, decision making, and structural modifications. Network modifications are performed by adapting existing weights such as to incur minimal changes to input-output behaviour.

Our framework for a combined ANN training and neural architecture search consists of three main components: a module that can perform a number of minimally invasive network operations ("surgeries"), a module that analyses network weights and can give recommendations which modifications are most likely to increase (validation) accuracy, and finally a module that serves as a genetic algorithm (the "Surgeon"), containing the former two while gradually evolving any given starting network. With the Surgeon, we are able to evolve and improve models for the SVHN (Netzer et al., 2011) and CIFAR-10 (Krizhevsky, 2009) datasets for varying starting topologies. We achieve particularly good results on starting topologies that would a posteriori have proven to be suboptimal. A great benefit of our approach is that it adds topology optimization to the ML training at a very limited additional computational cost. Convergence is reached for all test cases within a few CPU hours.

This paper contributes a computationally cheap ansatz for a genetic neural architecture search algorithm that makes evolutionary decisions based on mathematical analysis.

## 2  RELATED WORK

Neural architecture search (NAS) has been an increasingly popular research topic for many years (Elsken et al., 2019), starting as early as Miller et al. (1989), who presented one of the earliest neuro-evolutionary algorithms to search for suitable network topologies. Recent approaches by Dong & Yang (2019); Li & Talwalkar (2019), and Zoph & Le (2017) reach competitive performance on benchmark datasets such as CIFAR-10. However, this often comes at the cost of vast computational resources, with Zoph & Le (2017) making use of up to 800 GPUs for several weeks.

Cai et al. (2018) attempt to reduce computational costs by re-using network weights, as well as training and applying a reinforcement meta-controller for structural decisions. They make use of a number of function-preserving transformations (net2net) introduced by Chen et al. (2016), and extend them to allow also non-sequential network structures, such as DenseNet (Huang et al., 2017). DiMattina & Zhang (2010) introduce and rigorously prove conditions, under which gradual changes of the parametrization of a neural network are possible, while keeping the input-output behaviour constant.

There are also a number of neural architecture search strategies that do not depend on manual network modifications. Dong & Yang (2019) represent the search space as a directed acyclic graph and propose an efficient search algorithm. İrsoy & Alpaydın (2020) learn the network structure via so-called "budding perceptrons", in which an extra parameter is learned for each layer, that indicates whether or not any given node needs to branch out again or be removed altogether. Their method focuses on growing the network to the required size from a minimal starting topology. Frankle & Carbin (2019) present a method to identify particularly good network initializations that can train sparse networks to competitive accuracy. Another approach in NAS is to prune down from a larger starting topology (Blalock et al., 2020). Popular pruning techniques include applying SVD to existing network weights (Girshick, 2015; Denton et al., 2014; Xue et al., 2013).

The novelty of our research lies in combining existing tools such as net2net (Chen et al., 2016) and SVD with a genetic algorithm that modifies the given network in a decision based process instead of utilizing a black-box like decision module. To our best knowledge no such method has yet been proposed.

## 3  METHODS

This work introduces and utilizes three main modules:

- modification module: performs network modifications ("surgeries") so as to incur minimal changes to input-output behaviour.

- recommendation module: analyzes network weights and gives recommendations on which operations are most likely to improve network accuracy.

- "the Surgeon": a genetic algorithm that links the above two modules, and gradually evolves a given starting network.

### 3.1 THE MODIFICATION MODULE

We need to be able to perform a number of different modifications that can restructure network architecture. Ideally, these operations should not change at all the input-output behaviour. In cases where this is not possible, we aim to perform minimal impact changes instead. We perform four different types of modifications: adding or removing neurons or whole layers.

**Adding neurons and layers.** Under piecewise linear activation functions such as ReLu, one can always add neurons to a hidden layer, or even add whole layers, without any change to the overall network behaviour. Adding a whole layer is done by using an identity matrix as a new weight matrix for this inserted layer. We add a new neuron to any layer by dividing all weights for one existing neuron by two, and duplicating the respective connections in the next layer (cf. appendix A.1).

**Removing neurons.** Removing neurons from a layer is rarely possible without changing the input-output behavior unless some of the units are degenerate to begin with, i.e. the weight matrix is of reduced rank. For a modification with minimal impact on input-output behaviour we need the closest[1] possible projection onto a lower rank subspace. The Eckart-Young(-Mirsky) theorem states that this projection can be found by applying Singular Value Decomposition (SVD) to the weight matrix:

Let $A \in \mathbb{R}^{m \times n}$ be the (weight) matrix of some layer of interest in a neural network, then there exists a representation

$$A = U \Sigma V^T \tag{1}$$

with orthogonal $U \in \mathbb{R}^{m \times m}, V \in \mathbb{R}^{n \times n}$ and rectangular diagonal $\Sigma \in \mathbb{R}^{m \times n}$ with $k \leq \min(m, n)$ non-negative real entries $\sigma_i$ along its diagonal, called singular values of $A$ (cf. appendix A.2). In order to reduce the rank of $A$ to $r < k$, we set $\sigma_i = 0 \ \forall i > r$, and drop associated columns/rows of $U$ and $V$, which is feasible since by definition $\sigma_i \geq \sigma_{i+1}$. We then project to a smaller matrix $\tilde{A}$ by computing $U\Sigma$. To project back to the original layer size, so that the layer weight shapes match again, we multiply the weights of the subsequent layer with $V^T$.

Projecting onto a lower rank subspace by setting singular values to zero is sometimes called truncated SVD, and is used in network pruning (Denton et al., 2014; Girshick, 2015; Xue et al., 2013). This technique adds an intermediate layer of (potentially much fewer) neurons, which in case of very large weight matrices can drastically reduce the overall connection count.

Note that this projection method is not concerned at all with a potential activation function after the network's modified layer. In particular, changing a layer's activation function is in general not possible without changes to input/output behaviour, since activation functions are usually non-linear. We therefore do not perform any additional modifications to counterbalance a change of activation.

**Removing layers.** We remove whole layers in a similar fashion, by multiplying the layer's weights onto the weights of the next layer. This again ignores any activation function between the two layers, and will thus cause a change in input-output behaviour.

**Recuperation from surgery.** As we have seen, network modifications will in general cause a small change in input-output behaviour, typically leading to a loss in prediction accuracy. In order to make up for this, all network modifications are given a small amount of recuperative training (several batches) before any comparison is made. The retraining amount was determined by trials and statistical analysis based on a dataset[2] which we do not otherwise use in our results, to avoid overfitting the search algorithm to a specific dataset.

### 3.2 THE RECOMMENDATION MODULE

For the decision *when* to execute *which* network modification, we perform a two-step analysis. As a first order criterion, we compute the amount of information carried by each neuron by looking at the layer's singular values. The number of neurons that may be removed from a layer depends on the count of singular values that are (close to) zero, or are several orders of magnitude smaller than the layer's largest singular value:

$$\text{removable neurons} = \{\sigma_i, \ i = 1 \ldots k \mid \sigma_i < \epsilon_1 \sigma_1 \vee \sigma_i < \epsilon_2, \ \epsilon_1, \ \epsilon_2 \in \mathbb{R}^+\} \tag{2}$$

---

[1]under Frobenius norm $\| \cdot \|_F$

[2]MNIST, (LeCun et al., 1998)

The number of removable neurons in relation to the layer's total neuron count gives an indication whether addition or removal operations are preferable (cf. appendix A.3).

The second order decision basis is the Bayesian information criterion (BIC). It weighs the number of model parameters and training samples against the likelihood function of predicted vs. true target data. Since all operations are chosen to incur minimal changes to predicted values, the difference in the likelihood function between two modifications becomes negligible compared to the number of parameters, which allows us to reduce the formula to the parameter count of the modified network (cf. appendix A.4).

Thus, all potential network modifications are ranked by two parameters. The Surgeon has two modes: it can either pick the top $n$ ranking operations per decision step (with $n$ being a tunable hyperparameter), or select the highest ranking operation of each type. Our experiments are performed with the latter option.

### 3.3 "THE SURGEON": A GENETIC ALGORITHM FOR NEURAL ARCHITECTURE SEARCH

The final module is the Surgeon, the genetic algorithm that searches for an optimized network architecture while training network weights at the same time. In this first paper about our new ansatz, we limit our focus on perhaps the most ubiquitous type of architecture: sequential networks (i.e. without recurrent or skip connections) consisting only of fully connected layers. As Cai et al. (2018) have shown, however, a number of the above described tools can be generalized easily also to non-sequential networks as well as convolutional layers.

---

**Algorithm 1** The Surgeon

1: **function** PERFORMSURGERY($model$)
2:     train model $[m]$ for initial number of epochs
3:     current branches = $[m]$
4:     **while** termination criteria not met **do**                    ▷ e.g. max. number of epochs
5:         **for** all current branches **do**
6:             determine potential modification candidates $[m_{cand}]$
7:         **repeat**
8:             select $[m_{sel}]$ from $[m_{cand}]$                    ▷ e.g. 7 competitors
9:             re-train $[m_{sel}]$ for small number of batches                    ▷ e.g. 15 batches
10:             compare $[m_{sel}]$, keep $n$ top scoring as new current branches ▷ e.g. 2 conc. branches
11:             train current branches to full epoch step                    ▷ e.g. 10 epochs
12:         **until** improvement achieved or max re-tries reached
13:     select final best scoring branch
14:     **return** optimized network $m_{opt}$

---

The overall structure of the Surgeon can be seen in algorithm 1. First, the provided model is pre-trained for an initial number of epochs, then the list of current branches is initialized with it. The choice of how many branches at most are being kept concurrently is a hyperparameter setting, and has a great influence on the total computational cost of the Surgeon.

We then evolve and continuously update the list of concurrent branches until termination criteria, such as the maximal number of epochs (if limited by computational resources) or a minimum accuracy threshold (if the topology optimization is used without a computational resource bottleneck), are met. At each decision point, the recommendation module analyses all networks in the list of current branches, and ranks all potential modifications. From these, in line 8 of algorithm 1, we select the most promising candidates, and perform the selected operations using the modification module.

The generated candidates are re-trained for several batches, to make up for lost performance, and then scored according to their validation accuracy $a$, validation accuracy gain $\Delta a$, and parameter count increase $\Delta f$ as a fraction of its parent's parameter count:

$$s = a + \frac{\exp{(\Delta a)}}{\exp{(\Delta f)}}. \tag{3}$$

This composite score serves to reduce greediness of the Surgeon, and is adapted from Cai et al. (2018) (cf. appendix A.5). The top scoring candidates are then kept as new concurrent branches, and trained up to the full epoch step.

As an option to alleviate greediness, the Surgeon can keep a one cycle memory. In this case, the newly selected concurrent branches are compared to previously best scoring ones, and retained only if their achieved accuracy is at least as good as the previous value. Should this not be the case, they are discarded and we backtrack one step. New potential candidates are provided by the recommendation module, and we train and compare these. Finally the best scoring branch is returned as the optimized network.

## 4 EXPERIMENTS

### 4.1 DATA

We evaluate the performance of the Surgeon on two standard benchmark datasets: CIFAR-10, and SVHN. We pick three starting topologies that are described in table 1, and perform several runs of the Surgeon on each one. The small one is consciously chosen to be insufficient, to mimic a case where a model is unknowingly trained with an inadequate network architecture. We average our results over several runs and several random seeds, and compare to results achieved by simply training the starting topology for the same number of epochs. Detailed machine properties and hyperparameter settings are listed in appendix B.

Table 1: Starting topologies for the Surgeon, excluding input and output layers

| Name | Hidden layer count | Hidden layer sizes | Activation |
|--------|--------------------|--------------------|------------|
| Small | 1 | 10 | ReLu |
| Medium | 3 | 10 - 10 - 10 | ReLu |
| Large | 3 | 300 - 100 - 100 | ReLu |

**The CIFAR-10 dataset.** The Canadian Institute For Advanced Research (CIFAR)-10 dataset was first published by Krizhevsky (2009). It contains $60,000$ $32 \times 32$ color images that are evenly divided into ten classes. 10,000 of the images are set aside for validation purposes. Following related works (Cai et al., 2018; Huang et al., 2017), we perform data augmentation by flipping images horizontally, as well as translating by 4 pixels in either up, down, left or right direction. This yields a dataset of 360,000 images. We then normalize these images using channel means and standard deviation.

**The SVHN dataset.** The (Google) Street View House Number (SVHN) dataset was first published by Netzer et al. (2011). It contains 73,257 training, as well as 26,032 validation color images. We use the cropped version, where images are of size $32 \times 32$, and fall into 10 classes according to the numbers 0-9. Additionally, 531,131 extra images of lower difficulty are available but currently not used. We do not apply data augmentation, but divide by 255 in order to normalize the data.

### 4.2 RESULTS

We apply the Surgeon to each combination of the above datasets and starting topologies (cf table 1). We do so a total of 15 times, re-initializing the numpy and tensorflow modules with a new random seed after every 3 runs, and report average statistics (cf table 2). As a baseline, we train the starting topology for the same number of epochs with each random seed. Note that throughout the entire section, unless otherwise stated, we report achieved validation accuracies rather than training accuracies.

To avoid overfitting on any specific dataset, we fix some hyperparameters for training (such as batch size, optimizer, and learning rate) before starting any runs with the Surgeon (cf. appendix B). No additional fine tuning is performed on any model.

**Overall Surgeon performance.** Table 2 and figure 1 both show an overview of average Surgeon performance for all starting topologies and datasets. We can see that in all cases, the Surgeon reaches

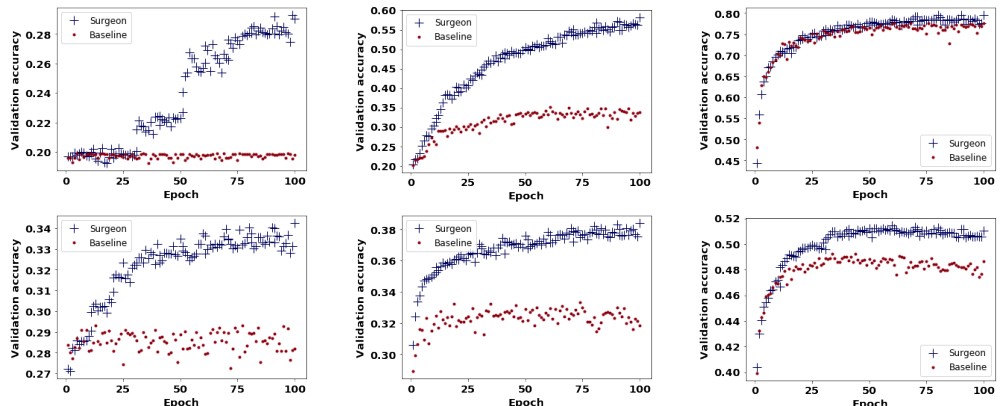

Figure 1: Average performance of the Surgeon over all 15 runs on the SVHN (top row) and CIFAR-10 (bottom row) datasets for small (left), medium (center), and large (right) starting topology, compared to training the starting topologies as a baseline.

or outperforms the result of the baseline. As expected, achieved accuracies for the SVHN dataset are higher than for the (almost 5 times larger) CIFAR-10 dataset.

In case of the SVHN dataset (figure 1 top row), the baseline for the small starting topology is not able to learn at all, whereas the only slightly larger medium topology is just about sufficient for at least a little bit of learning. The Surgeon is able to improve training in both cases, reaching 58% accuracy in case of the medium topology. At first glance it may seem peculiar that the Surgeon on average is able to reach only around 29% accuracy for the small one, given how similar the two of them are. This is due to averaging over a bi-modal distribution of results, where those branches that could be rescued (27 %) reach an average accuracy of 54% (in line with the medium topology), whereas the rest remain on baseline level (cf. table 2).

The large topology trained on the SVHN dataset yields similar accuracies with or without the aid of the Surgeon. What we can not see in figure 2 (top right) however, is that the Surgeon on average is able to reduce the total number of trainable parameters by around 7%, with an std of around 32% (cf. table 2 and figure 3 right).

For the CIFAR-10 set (figure 1 bottom row), both the small and medium starting topologies are too small for good results, even with the aid of the Surgeon, achieving an average of 34% and 36% accuracy respectively. On the large starting topology, the Surgeon is able to achieve on average at least a little improvement compared to the baseline. Parameter count is increasing in all three cases, which is indicative of the fact that neither of the chosen topologies is sufficient for a dataset of this size.

We will subsequently highlight a few interesting cases.

**Topology rescue.** We recall that the small starting topology, consisting of only one hidden layer with 10 neurons, was purposefully chosen to be insufficient for convergence. In fact, as we can see in the left column of figure 1, the baseline learns for neither SVHN nor CIFAR-10. The Surgeon on average manages to improve the topology and learn at least a little, though the achieved validation accuracy of around 29% and 34% respectively seems not great. In the case of SVHN however, the global average over all runs contains several instances where even with the Surgeon, the model is not able to learn at all and stays stuck in the initial state, as well as a number of runs where the Surgeon is able to very quickly leave this local sink and then in fact provides a model that learns very well.

In figure 2 (left), we can see a single run of the Surgeon, where an early *Add Layer* operation allows the network to train properly. The total parameter increase in this case is less than 2%, with the Surgeon preferring to add several small layers rather than widening existing layers. Note that due to the shape of our training data and choice of starting topology, a large portion of the network's parameters is required to connect the input layer to the first hidden layer. Adding a whole layer after the 10-unit layer causes only a small overall increase, whereas widening the first hidden layer might

Table 2: Statistics over all baseline and Surgeon runs. We report means and standard deviations.

| Topology | small | medium | large |
|---|---|---|---|
| **SVHN** | | | |
| Baseline accuracy | $0.20 \pm 0.00$ | $0.34 \pm 0.12$ | $0.78 \pm 0.01$ |
| Surgeon accuracy | $0.29 \pm 0.16$ | $0.58 \pm 0.04$ | $0.79 \pm 0.01$ |
| Relative accuracy increase [%] | 45.00 | 70.59 | 1.28 |
| Parameter fraction increase [%] | $16.48 \pm 52.0$ | $1.97 \pm 7.13$ | $-7.23 \pm 32.48$ |
| **CIFAR-10** | | | |
| Baseline accuracy | $0.28 \pm 0.01$ | $0.32 \pm 0.01$ | $0.49 \pm 0.00$ |
| Surgeon accuracy | $0.34 \pm 0.01$ | $0.38 \pm 0.06$ | $0.51 \pm 0.00$ |
| Relative accuracy increase [%] | 21.43 | 18.75 | 4.08 |
| Parameter fraction increase [%] | $1.12 \pm 2.85$ | $203.87 \pm 417.45$ | $2.12 \pm 3.13$ |

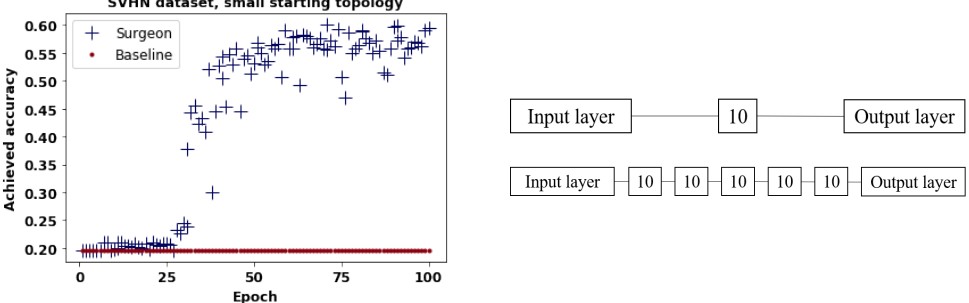

Figure 2: Left: Single run of the Surgeon on the SVHN dataset and small starting topology. Right top: Small starting topology. Right bottom: Topology after 100 epochs training with the Surgeon.

cause a greater increment in parameter count. Figure 2 (right) shows the starting topology (top) and the result after training with the Surgeon for 100 epochs (bottom). Note that up until epoch 50, only two layers have been added. In particular in this case, an earlier stopping point would have yielded similar performance with a lower parameter count.

**Accuracy increase.** The Surgeon is able to detect and improve sub-optimally sized network architectures. This works in cases such as above (figure 2 left), where it is very obvious that little to no learning happens, but also in less apparent ones, where the base topology does learn to a certain extent. In figure 3 (left) we can see a single run of the Surgeon trained on CIFAR-10 where both the large starting topology as well as the Surgeon start learning in a similar fashion and soon reach a plateau. After a while however, the Surgeon is able to perform an *Add Layer* operation that allows the topology to overcome the local optimum which had been reached.

**Parameter reduction.** As mentioned above, using the large topology as a starting point for the Surgeon does not improve achieved accuracy by any large margin (figure 2, right column). The baseline in this case is already performing quite well given that we are regarding a very basic network architecture. In this case we can observe pruning by the Surgeon. The composite scoring function given in eq. 3 prevents any big jumps in accuracy since they would most likely come at the cost of a (potentially rather large) increase in network size. The scoring function much rather prefers network operations that keep the accuracy more or less constant while reducing network size. On average, for the SVHN dataset, we are able to reduce the network's parameter count by around 7%. In figure 3 (right) we see an example where with an early *Remove Layer* operation, the Surgeon is able to reduce the total parameter count by almost 66%. The resulting loss in validation accuracy is compensated within just a few epochs of training.

**Computational costs.** All of our experiments are performed on a standard office computer, without making use of any GPU acceleration (cf. appendix B for detailed specs). In our trials the Surgeon

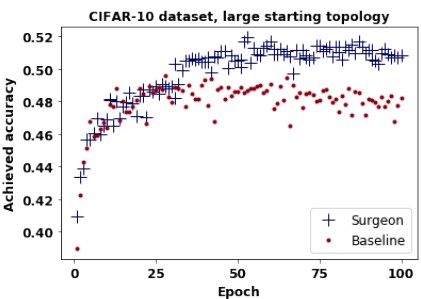
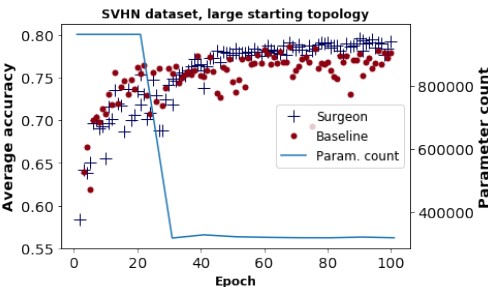

Figure 3: Left: Single run of the CIFAR-10 dataset and large starting topology. The Surgeon manages to overcome the local optimum at around epoch 30. Right: Single run of the Surgeon on the SVHN dataset and large starting topology. The Surgeon is able to reduce the total parameter count by almost 2/3 while still reaching the same overall validation accuracy.

is configured to produce a resulting topology that has been trained for exactly 100 epochs, with decision points every 10 epochs. We allow for a maximum of two concurrent branches, as well as re-drawing from potential branches up to two times. On average, 0.66 re-draws are necessary per decision step, resulting in an average total training amount of around 310 epochs per run of the Surgeon.

We can see from figure 1 that 100 epochs seems to be longer than is necessary for the Surgeon to reach convergence. With appropriate early stopping techniques, and/or a more dynamic training schedule, the total training amount for the Surgeon could be considerably reduced, and the resulting model (including its trained weights) used either as is, or further fine tuned.

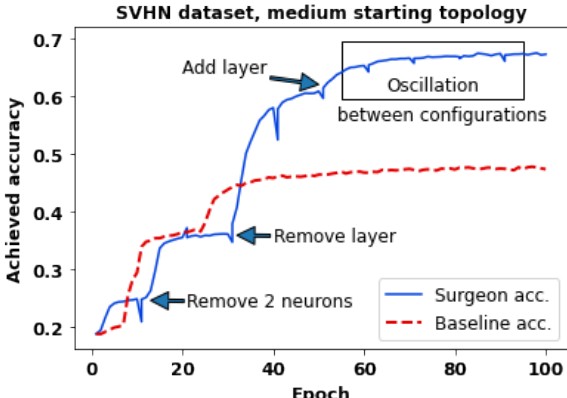

Figure 4: Example of a topology evolution performed by the Surgeon on the SVHN dataset and medium starting topology. Early on, reducing the network size helps to increase the accuracy level (at epochs 10 and 30). Adding a whole layer in epoch 50 is still able to achieve some increase in accuracy. From epoch 60 on the network oscillates between very similar states. Ideally this behaviour can be used as an indicator for early stopping in future versions of the Surgeon.

## 5  DISCUSSION AND OUTLOOK

In this paper, we presented the Surgeon, a ANN/evolutionary algorithm hybrid optimization designed for neural architecture search. The algorithm utilizes a modification module, that is able to perform minimally invasive network surgeries, where the topology of the network is modified with as little change to overall input-output-behaviour as possible. Additionally, it uses a recommendation module, that analyses a given neural network and indicates which structural changes may be most beneficial. Those changes can be either increases of network width or depth in case of high

information density, or respective decreases, in case network size can be reduced without fear of too high an accuracy loss. Both modules do not utilize any black box behaviour but are based on mathematical tools, which we presented in section 3.

We put the Surgeon to the test on several combinations of starting topologies and the SVHN and CIFAR-10 datasets. We saw that the network generated by the Surgeon is able to outperform the baseline in case of suboptimal topologies, or reach comparable accuracies while pruning the underlying network structure to less resource-intensive topologies.

A very important feature of the Surgeon is that it itself is computationally cheap, with little overhead compared to baseline training. Via hyperparameter settings, it is possible to make use of larger computational power if required/available. For this proof of concept work, we limited ourselves to the most basic and ubiquitous network structures - dense layers linked strictly in sequence.

For future work we wish to extend and improve both the Surgeon as well as the underlying modules. Currently, the Surgeon follows a static routine, with hyperparameters such as epochs steps, number of selected candidates, or number of concurrent branches staying constant throughout the whole process. This could be changed to a more dynamic approach, and could maybe integrate further features such as adaptive learning rates, or a more sophisticated memory, as well as early stopping mechanisms to improve performance gains compared to baseline training even further. The recommendation module can be improved by finding a closer approximation for the BIC.

Lastly we want to expand the modification module to allow more complex topologies as well, and include an option to cross architecture types. This would allow us to include e.g. convolutional or recurrent elements, change a network from one type to another, or even freely mix and match as required.

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

## A  APPENDIX A - MATHEMATICAL BACKGROUND

### A.1  ADDING LAYERS AND NEURONS WITHOUT CHANGE TO INPUT/OUTPUT BEHAVIOUR UNDER PIECEWISE LINEAR ACTIVATION FUNCTIONS

Adding whole layers to feed forward neural networks without change to the network's input-output behaviour is possible, if and only if the layer's activation function is at least piecewise linear (Chen et al., 2016). The number $n$ of units in the given layer has to be identical to the number of units in the following layer. We initialize the weights as an identity matrix $\boldsymbol{I} \in \mathbb{R}^n$. Let $\Phi$ denote the activation function of a dense layer and $\boldsymbol{x}$ an arbitrary input, then

$$\Phi(\boldsymbol{x}) = \Phi(\boldsymbol{I}\Phi(\boldsymbol{x})) \tag{4}$$

if and only if $\Phi$ is at least piecewise linear. In particular, for ReLu activation, adding layers without changing the input-output behaviour is possible.

$\square$

For adding neurons to an existing layer, consider the following example. Let $\boldsymbol{x} \in \mathbb{R}^n$ be the input to a layer with weights $\boldsymbol{A} \in \mathbb{R}^{m \times n}$, and $\boldsymbol{B} \in \mathbb{R}^{k \times m}$ the next layer's weights. Assume the activation

function $\Phi$ acting between the two layers to be an identity mapping. Then the output $\boldsymbol{y} \in \mathbb{R}^k$ is given by

$$\boldsymbol{y} = \boldsymbol{B} \cdot \Phi(\boldsymbol{Ax}) = \boldsymbol{B} \cdot (\boldsymbol{Ax}) \tag{5}$$

In particular, the $j^{th}$ entry of $\boldsymbol{y}$ is

$$\boldsymbol{y}_j = (b_{j1}, b_{j2}, \ldots, b_{jm}) \cdot \begin{bmatrix} \sum_{i=1}^n x_i a_{1i} \\ \sum_{i=1}^n x_i a_{2i} \\ \vdots \\ \sum_{i=1}^n x_i a_{mi} \end{bmatrix} \tag{6}$$

$$= b_{j1} \sum_{i=1}^n x_i a_{1i} + \cdots + b_{jm} \sum_{i=1}^n x_i a_{1m}, \tag{7}$$

where $a_{ij}$ is the $ij$ element of $\boldsymbol{A}$, and $b_{ij}$ the $ij$ element of $\boldsymbol{B}$. We now arbitrarily pick unit $m$ and duplicate its incoming and outgoing weights. Note that we also have to divide by 2 in order to keep the total sum constant.

$$\boldsymbol{y}_j = b_{j1} \sum_{i=1}^n x_i a_{1i} + \cdots + \frac{b_{jm}}{2} \sum_{i=1}^n x_i a_{1m} + \frac{b_{jm}}{2} \sum_{i=1}^n x_i a_{1m} \tag{8}$$

$$= (b_{j1}, b_{j2}, \ldots, \frac{b_{jm}}{2}, \frac{b_{jm}}{2}) \cdot \begin{bmatrix} \sum_{i=1}^n x_i a_{1i} \\ \sum_{i=1}^n x_i a_{2i} \\ \vdots \\ \sum_{i=1}^n x_i a_{mi} \\ \sum_{i=1}^n x_i a_{mi} \end{bmatrix} \tag{9}$$

This methods yields the exact same output $\boldsymbol{y} \in \mathbb{R}^k$ given input $\boldsymbol{x} \in \mathbb{R}^n$, but the weight matrices $\boldsymbol{A}$ and $\boldsymbol{B}$ are now of dimension $(m+1, n)$ and $(k, m+1)$ respectively.

Recall now that we chose the activation $\Phi$ to be the identity mapping. For any other activation function, the equation in line (8) holds true, if and only if this activation is at least piecewise linear.

$\square$

Chen et al. (2016) base their Net2WiderNet and Net2DeeperNet transformation on these steps.

## A.2  SINGULAR VALUE DECOMPOSITION AND PROJECTION ON LOWER RANK SUBSPACES

Removing units from a neural network's dense layer can never be done completely without change to input-output behaviour, unless the weight matrix is of reduced rank to begin with. Assuming that this is not the case, we need a method to generate the closest possible lower rank representation of our weight matrix, and find a projection onto it. Here we introduce such a method and specify what we mean by closest possible.

**Singular value decomposition.** Let $\boldsymbol{A} \in \mathbb{R}^{m \times n}$ be an arbitrary, real-valued matrix of rank $k \leq \min(m, n)$. Then there exist *orthogonal*[3] matrices $\boldsymbol{U} \in \mathbb{R}^{m \times m}$, $\boldsymbol{V} \in \mathbb{R}^{n \times n}$, as well as a rectangular diagonal matrix $\Sigma \in \mathbb{R}^{m \times n}$ with $k$ non-negative real entries along its diagonal, such that

$$\boldsymbol{A} = \boldsymbol{U}\Sigma\boldsymbol{V}^T. \tag{10}$$

This representation is called the *Singular Value Decomposition* of $\boldsymbol{A}$. The non-negative diagonal entries $\sigma_i$ of $\Sigma$ are called *singular values* of $\boldsymbol{A}$, and are usually given in descending order, i.e. $\sigma_1 \geq \sigma_2 \geq \cdots \geq \sigma_k > 0$.

For some applications, a slightly different notation is chosen by dropping any rows and columns of $\Sigma$ that consist entirely of zeros. This yields a quadratic diagonal matrix $\Sigma \in \mathbb{R}^{k \times k}$ with $k$ non-negative diagonal entries. $\boldsymbol{U} \in \mathbb{R}^{m \times k}$ and $\boldsymbol{V} \in \mathbb{R}^{n \times k}$ are cropped accordingly by dropping the respective columns.

---

[3]$\boldsymbol{U}^T\boldsymbol{U} = \boldsymbol{U}\boldsymbol{U}^T = \boldsymbol{I}$

**A measure for closeness.** Let further $|| \cdot ||_F$ denote the *Frobenius norm*

$$||\boldsymbol{A}||_F^2 = \sum_{i=1}^{m} \sum_{j=1}^{n} |a_{ij}|^2 = \text{trace}(\boldsymbol{A}^T \boldsymbol{A}). \tag{11}$$

It can be shown that $||\boldsymbol{A}||_F^2 = \sum \sigma_i^2(\boldsymbol{A})$, since $\sigma_i(\boldsymbol{A})$ corresponds to the square root of the $i^{th}$ non-zero eigenvalue of $\boldsymbol{A}^T \boldsymbol{A}$.

**The closest lower rank representation.** The matrices $\boldsymbol{U}$ and $\boldsymbol{V}$ act as projectors to and from a rank $k$ subspace of $\boldsymbol{A}$, spanned by the $k$ non-zero entries $\sigma_i$ of $\Sigma$. If we now wish to find a projection from $\boldsymbol{A}$ to a matrix of rank $r < k$, all we need to do is set the singular values $\sigma_{r+1}, \ldots, \sigma_k$ to zero, yielding a matrix $\boldsymbol{A}_r$ of rank $r$. The Eckart-Young(-Mirsky) theorem states that this matrix is the closest rank $r$ representation of $\boldsymbol{A}$ under the Frobenius norm, or

$$||\boldsymbol{A} - \boldsymbol{A}_r||_F \leq ||\boldsymbol{A} - \boldsymbol{B}||_F \ \forall \boldsymbol{B} \text{ of } \text{rank}(\boldsymbol{B}) = r. \tag{12}$$

This property is the core for a number of rank reduction techniques such as principal component analysis (PCA).

**Application for network modifications.** Think now of $\boldsymbol{A}$ as the weight matrix for a dense layer. As we already stated, the singular values $\sigma_i$ are always non-negative and ordered by descending magnitude. In particular, the magnitudes and magnitude distribution of the singular values allows us to infer how many units are actually needed to carry all of this layer's information.

This number may be surprisingly small compared to the number of units present, since (given a large enough dense layer) normally the first few singular values make up for the majority of contained information (Xue et al., 2013). A number of publications (Denton et al., 2014; Girshick, 2015) make use of this fact for various pruning techniques.

## A.3 ANALYTICAL CRITERIA FOR MODEL SELECTION

The recommendation module analyses each potential network topology modification, and scores it according to two criteria. For the first order decision criterion, the number of a layer's removable neurons is calculated as

$$n_r = |\{\sigma_i, \ i = 1 \ldots k \mid \sigma_i < \epsilon_1 \sigma_1 \vee \sigma_i < \epsilon_2, \ \epsilon_1, \ \epsilon_2 \in \mathbb{R}^+\}|. \tag{13}$$

This number is compared to the layer's total neuron count $n_l$. Let $c_l$ be the number of hidden layers in the network, and $l_i$ be the index of the layer in which the modification was performed. Then each modification candidate is given a score between 0 and 1, calculated as follows:

- add neurons: $1 - n_r/n_l$
- add layer: $(1 - n_r/n_t) \cdot l_i/c_l$
- remove neurons: $n_r/n_l$
- remove layer: 1, if $n_r = n_l$, otherwise $(n_r/n_l) \cdot l_i/c_l$

In particular, the higher the layer number, the more likely a layer is added or removed. For further refinement in the future, we aim to replace this selection criterion with a more sophisticated formula, which would improve the optimization process beyond the results presented herein.

The second order decision criterion is discussed in the next section.

## A.4 STATISTICAL CRITERIA FOR MODEL SELECTION

The Bayesian information criterion, also known as Schwarz information criterion (BIC), was derived by Schwarz (1978) to address the problem of selecting between (statistical) models of different dimension. It takes into account the number of parameters of the given model, the sample size of the input data, as well as the a-posteriori model error computed from a likelihood function of the model given its parameters and input data. Since methods from Bayesian statistics are applied, it is assumed that the underlying data are independent and identically distributed from a family of allowed distributions.

The Bayesian information criterion is given by

$$\text{BIC} = \ln n \cdot k - 2 \cdot \ln L, \tag{14}$$

where $k$ is the number of model parameters, $n$ the sample size, and $L$ the likelihood function.

In our application, sample size is constant throughout the whole process. $L$ needs to be estimated or calculated a-posteriori after a network operation has been performed. Our modifications are intended to keep the change in input-output behaviour minimal, therefore the difference in $L$ will be very small between any two modifications. Eq. (14) thus becomes

$$\text{BIC} \approx c_1 \cdot k + c_1 + \Delta L \tag{15}$$

where $c_1$ and $c_2$ are constant and $\Delta L$ is the (small) change in error depending on the performed operation. Since the number of parameters $k$ may be well above $10^6$ and $c_1 = \ln n \gg 1$ is dependent on the sample size, we neglect $\Delta L$ and the constants, and directly use $k$ as a second order constraint when deciding which modifications to apply.

## A.5 Evaluating Competing Branches

Our main objective function is to maximize the accuracy achieved by any given neural network through continuous network surgery. It is therefore sensible to retain those network candidates in algorithm 1, line 10 that reach the highest validation accuracy score. Note that we are scoring on validation accuracy instead of accuracy to avoid overfitting.

Focusing on accuracy alone is a greedy approach and carries the risk of getting stuck in local optima. We overcome this by additionally rewarding modification candidates that show a greater accuracy gain. However we need to make a distinction when rewarding this gain. We share Cai et al. (2018)'s rationale that an increase of 1% needs to be weighed higher in case it happens from 90 to 91% accuracy rather than 70 to 71%. At the same time, if an operation keeps the accuracy constant at e.g. 95% we can assume that a local optimum may have been reached. Therefore an operation that leads to an accuracy increase from 90% to 94%, showing potential for further improvement, should be regarded higher even though the reached accuracy is lower. Lastly we do not wish to neglect network size. A 1% accuracy increase might never be favourable at all, if the required increase in network size is "too big".

Note that it is hard to define when a network has indeed become "too big", a fact that is emphasized by the large number of publications dealing with pruning techniques (Blalock et al., 2020).

We need to find a way to balance these three components (accuracy, accuracy gain, network size), and to create a composite score by which we can rank the performance of current branches. As an additional restriction, the composite score should not depend on any global candidate statistics, nor do we want to set a global limit for network size. Therefore we cannot in any meaningful way regard the total number of parameters of a modification candidate, since we lack overall comparison. Instead, for each candidate, we store the network size as a fraction of the candidate's parent's network size. Thus, a network size fraction greater 1 indicates growth, a fraction smaller 1 indicates shrinking, and the identity operation yields a size fraction of exactly 1.

We want the scoring function as well as its first order derivative to be strictly increasing with accuracy gain, but decreasing with size fraction. This behaviour needs to hold even when accuracy gain becomes 0 or network size fraction becomes 1. The following scoring function fulfills all specified requirements:

$$s = a + \frac{\exp\left(\Delta a\right)}{\exp\left(\Delta f\right)} \tag{16}$$

where $a$ is the current accuracy, $\Delta a$ the current accuracy gain, and $\Delta f$ the current network size fraction.

Table 3: Scoring example. The layer number is given in reference to the hidden layers. Winning operations, that are kept as new concurrent branches, are indicated with an asterisk.

| Operation | Position | Accuracy | Acc. gain | Param. frac. | Score |
|---|---|---|---|---|---|
| Remove layer * | layer 5 | 0.6023 | 0.0274 | 0.9977 | 0.9812 |
| Identity * | - | 0.6016 | 0.0268 | 1.0000 | 0.9795 |
| Remove 1 Neuron | layer 5 | 0.5829 | 0.0081 | 0.9997 | 0.9539 |
| Add layer | before layer 4 | 0.5773 | 0.0025 | 1.0030 | 0.9450 |
| Add 20 Neurons | layer 2 | 0.5678 | -0.0070 | 2.6544 | 0.6377 |

## B  APPENDIX B - HYPERPARAMETER SETTINGS AND MACHINE SPECIFICATIONS

All our simulations were performed on a Windows 10 machine with an Intel(R) Core(TM) i7-9700K CPU 3.6GHz processor and 64,0 GB RAM.

All our code was implemented in python 3.7.7 using tensorflow 2.1.0. We chose the following hyperparameters:

**The modification module.**

- Amount of re-training per modification type:
    - Identity: $1 \cdot$ re-training batches
    - Add Neuron: $1 \cdot$ re-training batches
    - Add Layer: $10 \cdot$ re-training batches
    - Remove Neuron: $10 \cdot$ re-training batches
    - Remove Layer: $10 \cdot$ re-training batches
    - Truncated SVD: $1 \cdot$ re-training batches

**The recommendation module.**

- Absolute singular value threshold: $\epsilon_2 = 0.3$
- Relative singular value threshold: $\epsilon_1 = 0.005$
- Recommendation style: best scoring per modification type
- Include additional random draw: True

**The Surgeon.**

- Training time for winning branch: 100 Epochs
- Initial pre-training: 10 Epochs
- Interval between decision points: 10 Epochs
- Concurrent branches: 2
- Maximum re-draws per decision step: 2
- Number of batches for re-training: 25 batches

**The testing setup.**

- Random seeds: 6, 63, 72, 77, 97
- Loss function: Sparse categorical crossentropy
- Optimizer: SGD
    - learning rate = 0.005
- Metrics: accuracy
- Batch size: 16

