# OpenReview forum: "Neural Network Surgery: Combining Training with Topology Optimization"
_ICLR.cc/2021/Conference — Reject_

### Official Review · AnonReviewer1 · 2020-10-26
**suggest a rejection**

**Rating:** 4
**Confidence:** 4

**Review:**

Summary:
This study aims to search for topology jointly with the network training. The topology is optimized by a heuristic search in structural modifications including adding/removing neurons/layers. Experiments on SVHN and CIFAR-10 are conducted to show the effectiveness.

Pros:
The paper is well organized. The proposed method enjoys good interpretability because the decision of modification is made based on the weight matrix properties by SVD and projecting onto its lower-rank subspaces.

Cons:
1.	The major problem of this study is its extendibility. The method is only tested on fully-connected networks with very limited network depths and layer sizes. When the network is deep with much more non-linear activations, is the proposed method still valid? Besides, the authors only claim that the method can be generalized to convolutional network. But there is no corresponding description about how to generalize to convolutional network and its experimental results.
2.	The current experiments are not sufficient. The network and datasets used to test are too simple and the performance is far from being acceptable. It is weird to use fully-connected network on real-image datasets, CIFAR-10 and SVHN. Besides, the training time, computational and memory cost should also be reported and compared with the baseline method.
3.	The novelties are limited. The methods proposed in this paper are mainly heuristic implementations.

---

### Official Review · AnonReviewer2 · 2020-10-28
**Official Blind Review #2**

**Rating:** 4
**Confidence:** 4

**Review:**

The paper seems to devise a set of modification modules and develops an overall genetic algorithm that utilizes the recommendation module to identify optimized topologies for networks.

The paper is a combination of genetic algorithm with recommendation rule which builds on some tools like BIC. While these tools are organized in a reasonable manner, the overall idea seems rather simplistic and the contributions seem rather humble.

Importantly, the main concerns about the paper revolves around the evaluation. For example, the surgeon is tested on very small networks. As such, the evaluation seems very primitive. Without the results of networks for large datasets like ImageNet, it is hard to give credit to their evaluation nor gain confidence about the performance of this method in real scenarios.

Also, it would be very interesting to see the evolution of the topology. Maybe some figures may help the readers visualize the overall procedure. While this may be excessive for small networks like the ones authors rely on, I believe this may lead to new and interesting observations.

Furthermore, it would be very inspiring if the paper presents some prospect of the work in improving the mainstream neural architecture search algorithms. I believe such evaluation with some empirical results would make this a much more relevant paper for people working on improving neural architecture search.

---

> ### Comment · AnonReviewer2 · 2020-11-24
> **Comments**
>
> Answering the authors' question, Regarding mainstream NAS algorithms, there have been many including DARTS, ProxylessNAS. I believe if this paper can be interweaved with these works, it would be interesting. In fact, presenting ImageNet results in relation to such direction of work could be a better application of the technique presented in the paper.
>
> Having reviewed the revised manuscript and the authors' comments, as the work still lacks the results for ImageNet and yet hard to see how it can be applied to the current mainstream NAS, I will stay with the same score.

---

### Official Review · AnonReviewer3 · 2020-11-03
**Good framework, weak theoretical ground and experiments**

**Rating:** 5
**Confidence:** 2

**Review:**

This paper presents a neural network training framework based on a genetic algorithm called Surgeon. The idea consists of two modules. First, a series of network structure changes that aims to minimize the network input and output. Second, a heuristic (with regard to the objective function) for ranking and selecting the potentially good network modifications. The algorithm works iteratively between the former propose candidate network changes, and the later decides which candidate to accept and evolve into.

The paper is well written and easy to follow. As the authors stated, the paper is the first of the series of work. The framework presented is rigorous and flexible. I am fairly optimistic that the idea itself is pointing to a promising direction for exploring the neural network structure. However, I think the theoretical basis and experiments are a bit weak in their current form.

I have two main concerns. First, the heuristic used by Surgeon, either by weights or by BIC, does not establish direct association to the objective function that the network is optimizing. Therefore there is no any form of guarantee that the evolution will converge to some optima. Second, the experiments are initiated from rather simple baseline, without comparing against more sophisticated work or discussing why other state-of-the-art results are not relevant in this context.

Overall I like this paper, but for the two reasons listed above I think it may fall short to be accepted.

---

### Official Review · AnonReviewer4 · 2020-11-09
**novelty & experiment**

**Rating:** 4
**Confidence:** 4

**Review:**

this paper presents the Surgeon, a ANN/evolutionary algorithm hybrid optimization de- signed for neural architecture search. On SVHN and CIFAR-10, the network generated by the Surgeon is able to outperform the baseline in case of suboptimal topologies, or reach comparable accuracies while pruning the underlying network structure to less resource-intensive topologies. My major concern is the novelty, as the SVD technique and net2net techniques have all be proposed before. The experiment is not solid. The accuracy on Cifar is far from the state of the art. There is no large scale datasets. There's no performance comparison with related work, making the paper less convincing.

---

### Author Response · Authors · 2020-11-18
**To all reviewers**

We want to thank all the reviewers for the time and effort they dedicated to provide feedback for our manuscript, and are greatful for the insightful comments and valuable suggestions for improvements to our paper.
There have been concerns about the extendability and comparison to other state of the art results on larger datasets such as ImageNet.
We will address all concerns individually, but in summary want to clarify that this publication is intended as a proof of concept and as such does not aim to compete with state of the art accuracy levels.

Novelty of results:
We see the novelty of our work in the combination of several techniques. There are a number of examples in the literature making use of either SVD or net2net techniques, but to the best of our knowledge these have not been applied in combination as a genetic algorithm, that does not rely on a black-box decision module.
We have included an additional paragraph at the end of section 2 (Related Work) to stress this point more clearly.

Convergence to optimum:
We regard a proof of convergence as outside of the scope of this publication, but of course recognize its importance.

Representation of network evolution:
We tried to depict network evolution in figures 2 and 3, where we present the topology before and after applying the surgeon (fig 2), as well as the evolution of the number of network parameters over time (fig 3). We are aware that neither figure provides a comprehensive overview of the evolution over time.
We have included a new figure (4) similar to figure 3 (right), where we hope that some of the adaptations made by the surgeon are highlighted in a more tangible way.

Comparability with state of the art:
We are aware that the results we present fall short of any state of the art accuracy levels. This is in part due to the fact that current state of the art is reached by much more complex networks/methods, which often rely on huge amounts of training time and quite large network structures.
The results we present were all achieved with fixed hyperparameter settings and no additional tuning was performed, to avoid meta-overfitting.

ImageNet:
We agree that additional experiments on larger scale datasets such as ImageNet can allow for better comparability with state of the art results.
However we also feel that results derived from this dataset will still be somewhat limited by our current restriction to only fully connected networks.
Integrating more sophisticated structures such as convolutional layers goes beyond the scope of this publication, but will be included in a follow up publication.

Improving mainstream neural architecture search algorithms:
This is a very interesting idea. Do you have any specific algorithms in mind, that in your opinion might particularly benefit or are well suited?

We hope that we have sufficiently addressed all concerns and welcome further discussion.
Based on your recommendations we added another paragraph to section 2 (Related Work), as well as an additional figure (Fig. 4) to help understand the evolutionary steps performed by the Surgeon.

---

### Decision · Program_Chairs · 2021-01-07
**Final Decision**

**Decision:**

Reject

**Comment:**

This work proposes a framework to search for the topology of an artificial neural network jointly with the network training, via a genetic algorithm that can decide structural actions, such as addition or removal of neurons and layers. An extra heuristic based on Bayesian information criterion helps the optimization process decide on its decisions about the topology. They demonstrate improvements over baseline fully-connected networks on SVHN and (augmented) CIFAR-10.

Reviewers and myself agree that this is an interesting idea, and that the paper is easy to follow. While I may not agree that we need to achieve SOTA on these datasets, or see large scale ImageNet-type experiments for novel ideas, I agree with the reviewers, esp R1's point that the current experiments are not satisfactory to meet the bar for acceptance at ICLR.

CIFAR-10 and SVHN are well-established tasks, and showing baseline accuracy of 75%/48% on them respectively doesn't seem to do them justice, especially when most methods (even with low compute requirements) can get > 95% on both, for the past few years. For this work to be of interest to the broader community, it needs to be improved to incorporate at least respectable baselines on these small datasets, and perhaps be improved to work beyond fully connected networks.

At this stage, we need to see a revision of the method and see improvements before an acceptance decision can be made.